# Developmental Population Pharmacokinetics-Pharmacodynamics of Meropenem in Chinese Neonates and Young Infants: Dosing Recommendations for Late-Onset Sepsis

**DOI:** 10.3390/children9121998

**Published:** 2022-12-19

**Authors:** Yue-E Wu, Chen Kou, Xue Li, Bo-Hao Tang, Bu-Fan Yao, Guo-Xiang Hao, Yi Zheng, John van den Anker, Dian-Ping You, A-Dong Shen, Wei Zhao

**Affiliations:** 1Department of Clinical Pharmacy, Key Laboratory of Chemical Biology (Ministry of Education), School of Pharmaceutical Sciences, Cheeloo College of Medicine, Shandong University, Jinan 250012, China; 2Department of Neonatology, Beijing Obstetrics and Gynecology Hospital, Capital Medical University, Beijing 100020, China; 3Division of Clinical Pharmacology, Children’s National Medical Center, Washington, DC 20010, USA; 4Departments of Pediatrics, Pharmacology & Physiology, George Washington University, School of Medicine and Health Sciences, Washington, DC 20052, USA; 5Department of Paediatric Pharmacology and Pharmacometrics, University Children’s Hospital Basel, 4056 Basel, Switzerland; 6Pediatric Research Institute, Children’s Hospital of Hebei Province Affiliated to Hebei Medical University, Shijiazhuang 050000, China; 7Beijing Key Laboratory of Pediatric Respiratory Infection Diseases, Key Laboratory of Major Diseases in Children, Ministry of Education, National Clinical Research Center for Respiratory Diseases, National Key Discipline of Pediatrics (Capital Medical University), Beijing Pediatric Research Institute, Beijing Children’s Hospital, Capital Medical University, National Center for Children’s Health, Beijing 100045, China; 8Children’s Hospital Affiliated to Zhengzhou University, Henan Children’s Hospital, Zhengzhou Children’s Hospital, Zhengzhou 450018, China; 9NMPA Key Laboratory for Clinical Research and Evaluation of Innovative Drug, Qilu Hospital of Shandong University, Shandong University, Jinan 250012, China

**Keywords:** meropenem, population pharmacokinetics–pharmacodynamics, neonates, young infants

## Abstract

The pharmacokinetic (PK) studies of meropenem in Chinese newborns with late-onset sepsis (LOS) are still lacking. Causative pathogens of LOS and their susceptibility patterns in China differ from the data abroad. We, therefore, conducted a developmental population pharmacokinetic–pharmacodynamic analysis in Chinese newborns with the goal to optimize meropenem dosing regimens for LOS therapy. An opportunistic sampling strategy was used to collect meropenem samples, followed by model building and validation. A Monte Carlo simulation was performed to show the probability of target attainment (PTA) for various dosages. The information from 78 newborns (postmenstrual age: 27.4–46.1 weeks) was compiled and had a good fit to a 1-compartment model that had first order elimination. The median (range) values of estimated weight–normalized volume of distribution (V)and clearance (CL) were 0.60 (0.51–0.69) L/kg and 0.16 (0.04–0.51) L/h/kg, respectively. Covariate analysis revealed that postnatal age (PNA), gestational age (GA) and current weight (CW) were the most important factors in describing meropenem PK. Simulation results showed for LOS with a minimal inhibitory concentration (MIC) of 8 mg/L, the doses of 30 mg/kg 3 times daily (TID) as a 1-h infusion for newborns with GA ≤ 37 weeks and 40 mg/kg TID as a 3-h infusion for those with GA > 37 weeks were optimal, with PTA of 71.71% and 75.08%, respectively. In conclusion, we proposed an evidence-based dosing regimen of meropenem for LOS in Chinese newborns by using the population pharmacokinetic–pharmacodynamic analysis, based on domestic common pathogens and their susceptibility patterns.

## 1. Introduction

Neonatal sepsis is a worldwide burden with substantial morbidity. Late-onset sepsis (LOS), mainly occurring more than 72 h after birth, has a high mortality rate (5–15%) [1,2,3]. A meropenem dose of 10–40 mg/kg is commonly prescribed in neonates and young infants, although its use is not licensed for those under three months of age [4,5,6]. Thus, there is an urgent need for an evidence-based dosing regimen of meropenem for newborns to balance antibacterial efficacy against potential toxicity.

In clinical practice, the most common pathogens causing LOS, as well as their susceptibility patterns, show substantial discrepancies among regions [7]. The causative pathogens outside of China are relatively diverse and susceptible to meropenem with a lower minimal inhibitory concentration (MIC) breakpoint (<2 mg/L) [8]. In China, *Escherichia coli* (*E. coli*) and *Klebsiella pneumoniae* (*K. pneumoniae*) are the primary Gram-negative bacteria responsible for LOS, and the latter causes the most deaths [3,9,10,11]. The data from the Chinese antimicrobial resistance surveillance network show, between 2005 and 2017, that the proportion of meropenem-resistant *K. pneumoniae* increases from 2.9% to 24.0% [12]. This necessitates a higher MIC cutoff, and as a consequence, the dosing regimens need to be adjusted accordingly. However, pharmacokinetic (PK) studies of meropenem in the Chinese pediatric population are mainly conducted in older age groups with a large age span (0.1–14.4 years) [13]. Little information is available regarding neonates and young infants, who tend to have different drug volumes of distribution (V) and clearance (CL) as compared to older children [14]. Thus, a developmental population pharmacokinetic (PopPK) study specifically aimed at Chinese neonates and young infants is necessary.

As outlined above, our ultimate goal was to conduct a PopPK analysis of meropenem in Chinese neonates and young infants and then propose an evidence-based dosing regimen for LOS on the basis of local common clinical isolates and their susceptibility patterns.

## 2. Methods

### 2.1. Study Design

Designed as a prospective, open-label study, this PopPK study of meropenem was conducted in the Children’s Hospital of Hebei Province affiliated to Hebei Medical University and Beijing Children’s Hospital. Neonates (postmenstrual age (PMA) less than 48 weeks) were included if they had been diagnosed with bacterial infection and treated with meropenem. Neonates were excluded if they were participating in other clinical studies or were expected to survive for a shorter time than the treatment period. The study was approved by the ethics committee. Written informed consent was obtained.

### 2.2. Dosing Regimen and Pharmacokinetic Sampling

Meropenem (Dainippon Sumitomo Pharma, Osaka, Japan) was used intravenously and an opportunistic sampling strategy by using samples obtained from remaining blood after laboratory testing, which were part of routine clinical care, was used to collect samples from any day during the meropenem treatment [15], and infusion and sampling times were precisely recorded. The collected blood samples were centrifuged for 10 min at 1500× *g*, and then, plasma and cells were separated and stored in an ultra-low temperature refrigerator (−80 °C) for analysis.

### 2.3. Analytical Method of Meropenem

Meropenem concentrations were analyzed by high-performance liquid chromatography. The internal standard was metronidazole. The calibration curve ranged from 0.2 ug/mL (lower limit of quantification) to 50 μg/mL. Additionally, the intra-day and inter-day coefficients of variation of controls were 2.2% and 3.7%, respectively.

### 2.4. Population Pharmacokinetic Modeling of Meropenem

The nonlinear mixed effects modeling program (NONMEM, V7.4, Icon Development Solutions, San Antonio, TX, USA) was used for PK analysis. The PK parameters were estimated by using the first order conditional estimation (FOCE) method with interaction, of which the inter-individual variability was estimated via an exponential model. The model is expressed as below:(1)$$\theta_{i}=\theta_{mean}\times exp\left(\eta i\right)$$
where *θi* and *θ_mean_* are subject specific value and population typical value, respectively. Then, *ηi* means variability between subjects and is assumed to conform a normal distribution with variance ω2 and a mean of zero.

A forward and backward selection procedure was used for covariate analysis. Birth weight (BW), current weight (CW), postnatal age (PNA), gestational age (GA), postmenstrual age (PMA), albumin (ALB), urea nitrogen (BUN) and serum creatinine (CREA) were potential variables of PK parameters on the basis of the likelihood ratio test.

All covariates whose objective function value (OFV) drops exceeded 3.84 were added simultaneously to construct a full PopPK model. Then, each covariate was independently removed. The covariate was considered to be notably correlated with the PK parameter if the OFV increased to more than 6.635 and was finally retained in the final PopPK model.

### 2.5. Model Validation

Internal and external validation were conducted to evaluate the applicability and predictivity of the final PopPK model, using both statistical and graphical criteria.

#### 2.5.1. Internal Validation

Goodness-of-fit plots of observed (DV) versus individual prediction (IPRED), DV versus population prediction (PRED), conditional weighted residuals (CWRES) versus PRED and CWRES versus time were used for model diagnosis. A nonparametric bootstrap with re-sampling and replacement was also used to assess model performance and stability. The sampling was repeated 1000 times. The values of estimated parameters from the original data set were compared with those estimated from the bootstrap procedure. The whole process was completed using PsN (v2.30). One thousand datasets were simulated using the final model parameters. Normalized prediction distribution errors (NPDE) results were presented graphically as provided by the NPDE R package (v1.2), using the histogram and QQ-plot. The NPDE is expected to conform to the N (0, 1) distribution. In addition, a visual predictive check was performed using R/Xpose, and the output came from PsN in the pirana environment.

#### 2.5.2. External Validation

An external validation dataset was collected from The First Affiliated Hospital of Shandong First Medical University, Jinan, China with opportunistic samples and sufficient information concerning patients, including physiological and covariate data, dosing and sampling history.

Mean prediction error (MPE), as well as mean absolute prediction error (MAPE), was calculated (Equations (2) and (3)) to represent the bias of the model’s applicable and predictive performance [16]. In addition, the percentage of newborns whose MPE values were within ±20% and ±30% were calculated, respectively [17].
(2)$$MPE=\frac{1}{N}\sum \frac{IPRED_{i}-OBS_{i}}{OBS_{i}}\times 100\%$$
(3)$$MAPE=\frac{1}{N}\sum \left|\frac{IPRED_{i}-OBS_{i}}{OBS_{i}}\right|\times 100\%$$

### 2.6. Dosing Regimen Evaluation and Optimization

Meropenem features time-dependent bacterial killing [18], and the pharmacokinetic-pharmacodynamic effect was exposed by the probability of target attainment (PTA), which showed the percentage of patients with free meropenem concentrations (approximately 98% of the total) beyond the MIC for 70 percent of the dose interval (70% fT > MIC) [19]. The domestic pathogens responsible for LOS and their susceptibility patterns were also incorporated for dosing optimization. The standard MIC value of 8 mg/L was used for LOS (PNA > 3 days).

Then, Monte Carlo simulations were carried out 1000 times to evaluate the PTA of different dosing regimens, covering 20 to 60 mg/kg, a dosing frequency of twice (BID) and three times daily (TID), and 1 h to 5 h infusion time. The simulation cohort originally involved 78 newborns and was split into two subgroups according to GA (35 newborns with GA > 37 weeks and43 with GA ≤ 37 weeks).

## 3. Results

### 3.1. Study Population

Seventy-eight patients were included on the basic of the inclusion and exclusion criteria. Informed consent was obtained. The mean (SD) values of GA, PNA and CW of the 78 newborns were 35.0 (4.65) weeks, 17.6 (19.2) days and 2661 (927) grams, respectively. The characteristics are presented in Table 1.

### 3.2. Model Building

One hundred and ten concentrations of meropenem were obtained for modeling. The profile of concentration versus time since last dose is shown in Figure 1. A one-compartment model with first-order elimination fitted the data. The distribution (V) and clearance (CL) of meropenem were used to parameterize the model. Residual variability and inter-individual variability were best described by a proportional model and an exponential model, respectively, and then, V and CL were estimated.

By incorporating priori CW into the basic model (allometric coefficients of 1 for V and 0.75 for CL), the allometric size method was employed, causing a significant OFV drop of 15.1 points. GA and PNA were determined as the most crucial covariates on CL, causing an OFV drop of 27.5 units.

The parameter estimates of the final PopPK model are summarized in Table 2. The median (range) values of estimated weight–normalized V and CL at steady-state were 0.60 (0.51–0.69) L/kg and 0.16 (0.04–0.51) L/h/kg, respectively.

### 3.3. Model Evaluation

#### 3.3.1. Internal Validation

Internal validation showed that the goodness-of-fit of the model was acceptable. Predictions were unbiased in terms of population and individual concentrations (Figure 2A,B). No trends were found in the plots of CWRES versus time and PRED (Figure 2C,D). The median parameter estimates derived from the bootstrap were in accordance with their counterparts from the final PopPK model, which indicated that the model could re-determine the estimates of PK parameters and was stable (Table 2). NPDE histogram and distribution met well the theoretical N (0, 1) distribution and density (Figure 2E,F), indicating that the model was in accordance with the individual data. The results of visual predictive checks are presented in Figure 3, confirming that the model had good simulation properties.

#### 3.3.2. External Validation

The external validation set was related to an independent cohort involving 16 newborns. The mean (SD) GA, PNA and CW at the time of study were 31.2 (5.29) weeks, 24.5 (9.90) days and 1857.0 (939.76) grams, respectively. Then, the validation result showed that the mean MPE and MAPE were 0.26 and 0.32, with 78% of MPE within ±20% and 83% within ±30%, respectively.

### 3.4. Dosing Regimen Evaluation and Optimization

Meropenem dose optimization showed the PTA value. In terms of the clinical definition of premature and term newborns, the cutoff of 37 weeks for GA was selected to divide age group.

The PTA values of different dosing regimens and age groups with the MIC susceptibility breakpoint of 8 mg/L are depicted in Figure 4. For newborns with GA ≤ 37 weeks, the regimens of twice daily necessitated a high unit dose and a long infusion time. The results showed that 60 mg/kg as a 3-h infusion, 50 mg/kg as a 4-h infusion and 40 mg/kg as a 5-h infusion were available, with PTAs of 72.49%, 73.54% and 74.09%, respectively. When the frequency increased to TID, the unit dose of 30 mg/kg as a 1-h infusion was sufficient, with PTA of 71.71%. For newborns with GA > 37 weeks, the BID regimens seemed inappropriate due to the higher dose. The TID regimens, 50 mg/kg as a 2-h infusion and 40 mg/kg as a 3-h infusion, could easily attain the target, with PTAs of 72.50% and 75.08%, respectively.

## 4. Discussion

A meropenem PopPK study was conducted in Chinese neonates and young infants. By incorporating pharmacometrics along with an opportunistic sampling strategy, the PK parameters for meropenem in newborns were determined, making the prediction and description of drug disposition during real clinical care accessible.

We found a one-compartment model with first-order elimination had a better fit for the data. The estimated weight-normalized CL median (range) value was 0.16 (0.04–0.51) L/h/kg, which exhibited subtle differences from previous studies abroad covering a similar age range [8,14,20,21,22]. An obvious explanation came from the extent of the kidney maturation anatomically and functionally [23,24,25]. A former study conducted abroad showed PMA could improve model fit [8], but our data identified the combination of PNA and GA was superior to PMA alone. Furthermore, contrary to common belief, CREA did not have a crucial effect on CL.

Dosing optimization for antimicrobial therapy is hugely dependent on safety of the used antibacterial agent, the developmental pharmacokinetics–pharmacodynamics of that agent and the underlying microbiology [26]. Regarding safety, meropenem is well tolerated in neonates and young infants and could function as a promising agent in therapy for LOS [20,27]. When referring to pharmacodynamics, the property of time-dependent killing necessitates attention to the % fT > MIC [18]. Accepting adult values (40–50% T > MIC) could easily induce underexposure because of the immunocompromised status of newborns and increasing antibiotic resistance [2,28,29]. Recently, some relevant research abroad typically set this value as 60% or even much lower [8,14,22]. However, a higher value, 70% T > MIC, is essential and more conservative, which is in accordance with other studies conducted in China [19]. The simulation results were exposed by the PTA, which was defined as the proportion of patients achieving 70% T > MIC and expected to exceed 70%.

In terms of microbiology, rational meropenem dosing regimens for LOS should integrate the most common pathogens and their susceptibility patterns based on domestic real conditions. Some foreign research found that the Gram-negative pathogens causing LOS were susceptible to meropenem, and a relatively low MIC value (2 mg/L) was adequate, thereby proposing the dosing regimens (20 mg/kg, TID) accordingly [8]. However, domestic reality makes this value improper. In China, the predominant Gram-negative pathogens responsible for LOS are *E. coli* and *K. pneumoniae* [3,10]. Recent efforts suggest that *E. coli* is susceptible to carbapenem antibiotics in China, with a resistance rate less than 5%. However, the carbapenem resistance among *K. pneumoniae* increases rapidly and the annual isolation rate of K. pneumoniae also shows a steady increase [12]. Finally, we chose MIC value of 8 mg/L to conduct the dosing simulation for LOS.

Dosing optimization results showed the previously proposed dosage, 20 mg/kg TID, was underdosed for newborns in different GA groups, only leading to PTA of 34.07% (GA > 37 weeks) and 57.76% (GA > 37 weeks) (as 1-h infusion), respectively. Furthermore, the results also indicated that a lower unit dose needs an increased infusion time to ensure an optimal PTA value. To avoid the overuse of meropenem and maintain the stability of the drug in the meanwhile, we suggested that for newborns with GA ≤ 37 weeks, 30 mg/kg TID as a 1-h infusion could exert a better effect with a PTA of 71.71%. Then, for newborns with GA > 37 weeks, 40 mg/kg TID as a 3-h infusion could easily attain the desired target, with a PTA of 75.08%. Although meropenem has a favorable safety profile in pediatrics [30,31], the potential nephrotoxic effect can be monitored by using creatinine and other alternative indicators such as cystatin and BUN level when the proposed regimens are used.

One limitation of this study relates to the scarcity of exploring meropenem concentrations in cerebrospinal fluid. Furthermore, the regimen we proposed needs further evaluation for both efficacy and safety in prospective clinical practice in newborns.

## 5. Conclusions

The PopPK study of meropenem was conducted in Chinese newborns, and an evidence-based dosing regimen was elaborated for LOS (PNA > 2–3 days). The pharmacometric method together with modeling techniques showed CW, GA and PNA could serve as the significant covariates on meropenem PK in newborns. Furthermore, it is anticipated that meropenem for newborns with LOS should be dosed on the basis of GA.

## Figures and Tables

**Figure 1 children-09-01998-f001:**
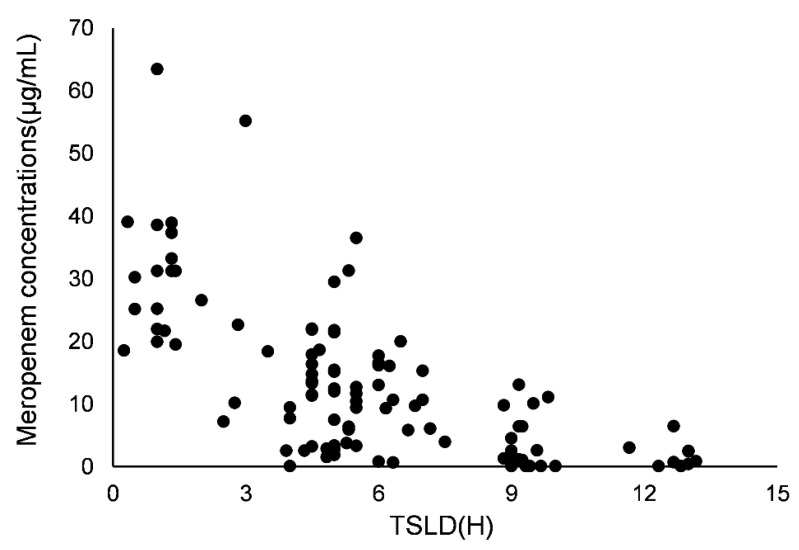
Meropenem concentrations versus time since last dose (TSLD).

**Figure 2 children-09-01998-f002:**
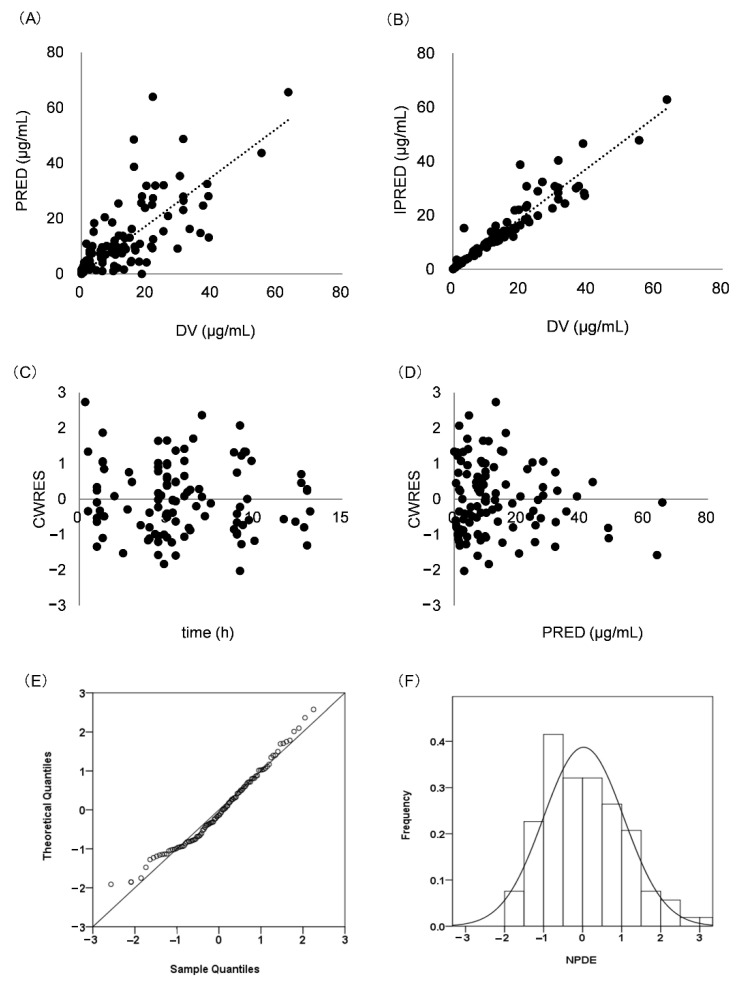
Model evaluation for meropenem. (**A**) PRED versus DV; (**B**) IPRED versus DV; (**C**) CWRES versus time; (**D**): CWRES versus PRED; (**E**) Histogram of the distribution of NPDE; (**F**) QQ-plot of the NPDE versus the theoretical N (0,1) distribution. Population prediction: PRED; observed values: DV; individual prediction: IPRED; conditional weighted residuals: CWRES; normalized prediction distribution errors: NPDE.

**Figure 3 children-09-01998-f003:**
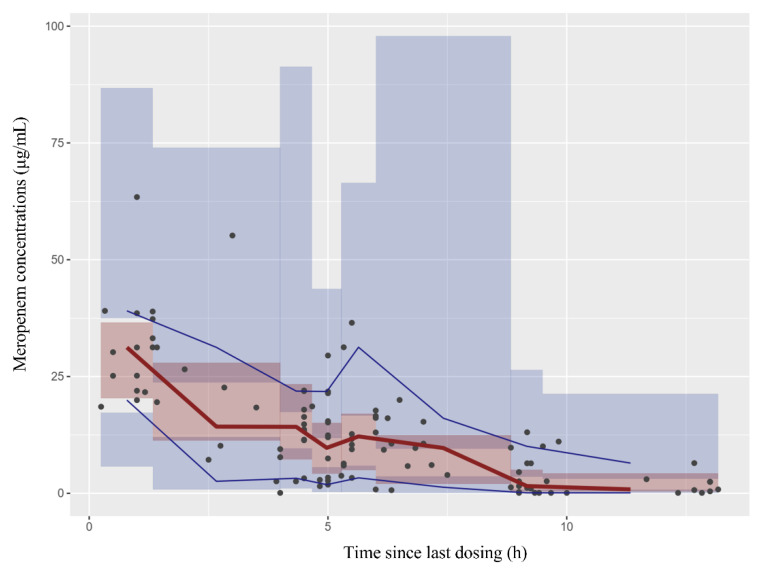
The visual predictive checks. The pink field represents simulation-based 95% confidence intervals for the median and the red line represents the median observed concentrations. The circles represent the observed concentrations. The 95% intervals for the model-predicted percentiles are shown as blue field and the observed 5th and 95th percentiles are indicated by blue lines.

**Figure 4 children-09-01998-f004:**
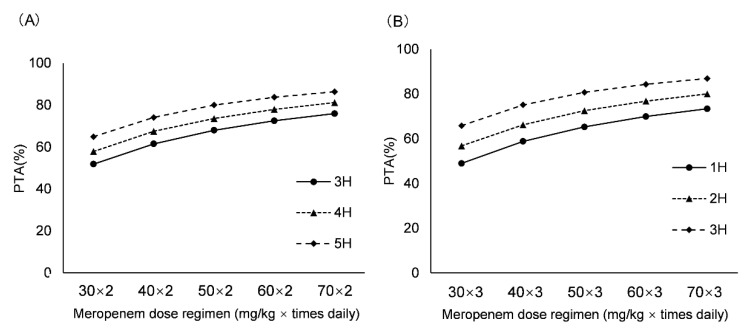
PTA of various dosing regimens for MIC value of 8 mg/L. (**A**) For newborns with GA ≤ 37 weeks; (**B**) For newborns with GA > 37 weeks. Probability of target attainment: PTA.

**Table 1 children-09-01998-t001:** Characteristics of 78 newborns.

	Mean (SD)	Median (Range)
PNA (days)	17.6 (19.2)	12.0 (1.00–113)
PMA (weeks)	37.5 (4.40)	38.2 (27.4–46.1)
GA (weeks)	35.0 (4.65)	36.5 (26.3–42.1)
BW (g)	2349 (973)	2500 (480–4100)
CW (g)	2661 (927)	2680 (980–5310)
CREA (µmol/L)	32.4 (27.1)	23.8 (1.1–145.2)
BUN (mmol/L)	4.4 (3.38)	3.79 (0.5–19.6)
ALB (g/L)	27.1 (5.84)	25.6 (16.7–44.6)
Dose (mg/dose)	57.6 (26.9)	55.0 (20.0–160)
Dose (mg/kg/dose)	21.8 (6.37)	19.7 (9.3–40.4)

Postnatal age: PNA; postmenstrual age: PMA; gestational age: GA; birth weight: BW; current weight: CW; serum creatinine: CREA; urea nitrogen: BUN; albumin: ALB.

**Table 2 children-09-01998-t002:** PopPK parameters and bootstrap results.

Parameters	Full Dataset		Bootstrap	
	Final Estimate	RSE(%)	Median	5th–95th
V (L)				
V = θ1 × (CW/2680)				
θ1	1.63	11.4	1.63	1.35–1.99
CL (L/h)				
CL = θ2 × (CW/2680)^0.75^ × F_age_				
θ2	0.503	7.90	0.522	0.439–0.605
F_age_ = (PNA/12)^θ3^ × (GA/36.5)^θ4^				
θ3	0.209	21.4	0.206	0.123–0.292
θ4	2.14	22.9	2.10	1.16–2.80
Inter-individual variability (%)				
V	17.5	58.8	23.7	9.28–55.0
CL	50.8	11.5	49.7	40.0–59.6
Residual variability (%)	28.8	15.5	24.9	15.2–34.4

Volume of distribution: V; clearance: CL; current weight in grams: CW; gestational age in weeks: GA; postnatal age in weeks: PNA. In our population, 2680 g, 36.5 weeks and 12 days are the median CW, GA and PNA values, respectively.

## Data Availability

The data that support the findings of this study are available from the corresponding author upon reasonable request.

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
