# Peer review of "Developmental Population Pharmacokinetics-Pharmacodynamics of Meropenem in Chinese Neonates and Young Infants: Dosing Recommendations for Late-Onset Sepsis"

_children, 2022, doi:10.3390/children9121998_

Round 1

Reviewer 1 Report

This is a good manuscript and should be published in public domain. 

My comments are as:

Details of exclusion criteria?

Details of timing and LOS day of meropenem blood sampling?

Reason for using albumin (ALB) as covariates however meropenems just binds only 2% with proteins?

Why do you select specifically 70%T>MIC instead of 40% or 100% as PD target in neonates’ population with LOS? could you please report for 100%T>MIC PD target?

Could you strongly justify allometric coefficients of 0.75 for CL and 1 for V in neonate population and why did not used alternative methods?

Do you justify the prediction (RSE is higher) of V in the model?

Could you please provide the VPC plots?

Thank You.

Author Response

Point 1: Details of exclusion criteria?

Response 1: We like to thank the reviewer for this comment. We rephrased the exclusion criteria: Neonates were excluded if they were participating in other clinical trials or had a expected survival time less than the treatment cycle.

Point 2: Details of timing and LOS day of meropenem blood sampling?

Response 2: We like to thank the reviewer for this comment. An opportunistic sampling strategy by using samples collected from blood remaining after routine laboratory tests as part of clinical care was selected to collect samples at any day during the meropenem treatment. The time of sampling and drug administration were accurately recorded. We added this information to the revised manuscript. From the final results, the range values of sampling times after last dose were 0.25-13.17 hours, which covered the dosing interval, ensuring the quality of sampling. The range of LOS day of meropenem blood sampling was 1-48 days.

Point 3: Reason for using albumin (ALB) as covariates however meropenems just binds only 2% with proteins?

Response 3: We like to thank the reviewer for this comment. We routinely tried each variable when we conducted the covariate analysis. It is true that ALB has no effect on the PK parameters of meropenem from a physiological point of view, and our results also confirm this.

Point 4: Why do you select specifically 70%T>MIC instead of 40% or 100% as PD target in neonates’ population with LOS? could you please report for 100%T>MIC PD target?

Response 4: We like to thank the reviewer for the comments. For adults, the recommended traditional fT>MIC values are 40% to 50%. However, neonates are considered immunocompromised, and they are more vulnerable to infection. In order to improve treatment efficacy and avoid the emergence of resistant bacteria, a higher fT>MIC target for effectiveness assessment is needed. A PK-PD study on β-lactam antibiotics showed an in vivo bacteriostatic effect for approximately 40% fT>MIC, whereas maximum killing was reached when 60%–70% fT>MIC. Thus, 70% fT>MIC was selected as the PD target, which was consistent with the value used in other studies of β-lactam antibiotics in neonates (Ther Drug Monit. 2012;34:312-9; Antimicrob Agents Chemother. 2019 Jan 29;63:e02336-18). In addition, we confirm that better efficacy can be achieved using the target of 100%. However, ADR monitoring data show that the adverse reactions rate of drugs in neonates is 4 times that in adults. Too much drug exposure may increase the risk of adverse reactions. Basing on a balance between maximal efficacy and minimal toxicity, as well as reduction of resistance, 70% T>MIC was selected as the PD target.

Point 5: Could you strongly justify allometric coefficients of 0.75 for CL and 1 for V in neonate population and why did not used alternative methods?

Response 5: We like to thank the reviewer for this comment. Indeed, there are different approaches to parameterizing weight effects in neonatal antimicrobial pharmacokinetic models. The point is to find the most proper way to quantify the effects of the covariate on the PK parameters. We tried both the fixed coefficient method and the estimated coefficient method, the fixed coefficient method was more stable. In addition, and the fixed coefficient is widely used in neonatal pharmacokinetic studies (Clin Pharmacol Ther. 2018;103:458-467; Antimicrob Agents Chemother. 2020;64:e01971-19; Sci Rep. 2021;11:6168; J Pediatr. 2011;159:414-419.e1-3).

Point 6: Do you justify the prediction (RSE is higher) of V in the model?

Response 6: We like to thank the reviewer for this comment. The RSE value in this study was a little high, which may be related to sparse blood collection. Due to ethical and operational issues in neonates, only sparse blood collection could be performed in this study. Although the RSE value is a little high, simulation-based evaluation (VPC, NPDE) are good in model validation.

Point 7: Could you please provide the VPC plots?

Response 7: We like to thank the reviewer for this suggestion. We added the VPC plots in the revised manuscript.

Reviewer 2 Report

Comments and sugestions:

In the abstract, at `conclusions` the authors repeat a similar phrase from the objectives, that should be deleted (row 44-45) and replaced with consistent conclusions regarding the study.

Did the authors notice any specific differences related to the type of Gram-negative germs?

A well designed and argumented study, therefore discussions should be improved by adding some more data regarding the potential nephrotoxic effect of meropenem in neonates. Although creatinine was not significantly altered, some remarks about BUN would improve the value of this work and could reassure the safety of the regimen proposed.

References should be revised, as, for example, ref 12 lacks pages or doi number. Please revise the consistency of references format for all the titles.

Conclusions should not refer to the limits of the study. These remarks should be included in Discussions session.

Author Response

Point 1: In the abstract, at `conclusions` the authors repeat a similar phrase from the objectives, that should be deleted (row 44-45) and replaced with consistent conclusions regarding the study.

Response 1: We like to thank the reviewer for this suggestion. We rephrased the sentence “In conclusion, we proposed an evidence-based dosing regimen of meropenem for LOS in Chinese neonates and young infants by using the population pharmacokinetic-pharmacodynamic analysis, based on domestic common pathogens and their susceptibility patterns” in the revised manuscript.

Point 2: Did the authors notice any specific differences related to the type of Gram-negative germs?

Response 2: We like to thank the reviewer for this comment. The common Gram-negative pathogens of LOS are mainly Escherichia coli and Klebsiella spp in China and other Asian countries, but their proportion is much higher than that in other regions and countries (Arch Dis Child Fetal Neonatal Ed. 2015;100:F257-63). In addition, foreign research found that the Gram-negative pathogens causing LOS were susceptible to meropenem and a relative low MIC value (2 mg/L) was adequate, which was different from the situation in China. Therefore, an evidence-based dosing regimen of meropenem for LOS in Chinese neonates and young infants should be made based on domestic common pathogens and their susceptibility patterns.

Point 3: A well designed and argumented study, therefore discussions should be improved by adding some more data regarding the potential nephrotoxic effect of meropenem in neonates. Although creatinine was not significantly altered, some remarks about BUN would improve the value of this work and could reassure the safety of the regimen proposed.

Response 3: We like to thank the reviewer for the comments. According to a review study in which the paediatric population base for which safety data are available has doubled to over 1000 patients, meropenem has a favourable safety profile, and the most common adverse events reported for meropenem were diarrhoea (2.5%), rash (1.4%) and nausea/vomiting (1.2%) (Drug Saf. 2007;30:657-68). In addition, no adverse events of nephrotoxicity were reported in a randomised controlled trial for meropenem conducted in the neonates with late onset sepsis (PLoS One. 2020;15:e0229380). The potential nephrotoxic effect of meropenem in neonates can be monitored by using the creatinine and other alternative indicators such as cystatin and BUN level when the proposed regimens are used. We added the information above to the discussion in the revised manuscript.

Point 4: References should be revised, as, for example, ref 12 lacks pages or doi number. Please revise the consistency of references format for all the titles.

Response 4: We like to thank the reviewer for this suggestion. We have added the pages of Ref 12, and checked the other references and unified the format in the revised manuscript.

Point 5: Conclusions should not refer to the limits of the study. These remarks should be included in Discussions session.

Response 5: We like to thank the reviewer for this suggestion. The limits of the study have been moved to the Discussions session in the revised manuscript.

Round 2

Reviewer 1 Report

Thanks, you very much for the updated version.